# Microbial Diversity and Community Variation in the Intestines of Layer Chickens

**DOI:** 10.3390/ani11030840

**Published:** 2021-03-16

**Authors:** Sha-Sha Xiao, Jian-Dui Mi, Liang Mei, Juanboo Liang, Kun-Xian Feng, Yin-Bao Wu, Xin-Di Liao, Yan Wang

**Affiliations:** 1National Engineering Research Center for Breeding Swine Industry, College of Animal Science, South China Agricultural University, Wushan Road, Tianhe District, Guangzhou 510642, China; 17727616099@163.com (S.-S.X.); mijiandui@163.com (J.-D.M.); 13600469985@163.com (L.M.); 13422174080@163.com (K.-X.F.); wuyinbao@scau.edu.cn (Y.-B.W.); xdliao@scau.edu.cn (X.-D.L.); 2Institute of Tropical Agriculture and Food Security, Universiti Putra Malaysia, Serdang 43400, Malaysia; jbliang@upm.edu.my

**Keywords:** white Lohmann layer chickens, intestinal microbiome, diversity and community, 16S rRNA

## Abstract

**Simple Summary:**

Early life is a critical window period for the colonization of intestinal microbiota in animals. The colonized intestinal microbiota during this early stage has an important influence on the growth of animals and the development of the immune system. Currently, little is known about how the layer chickens microbiome varies in different intestinal segments in the early life. This study determined the diversity and community variations in the duodenum, caecum and colorectum of white Lohmann layer chickens on fourteen different time points (0, 1, 3, 5, 7, 12, 18, 24, 30, 36, 40, 43, 50 and 57 days) by applying 16s rRNA sequencing. Our study found that the intestinal microbiota of white Lohmann layer chickens matured at day 50. In addition, the caecum and colorectum succession pattern is similar but different from that of the duodenum. When the intestinal microbiota matures, the dominant microorganisms in the duodenal intestine are *lactobacillus*, while the dominant microorganisms in the cecum and colorectum are more complex, mainly *Bacteroides*, *Odoribacter*, and *Clostridiales vadin BB60 group*. This study provides information about changes in the microbiota composition of layer hens with age.

**Abstract:**

The intestinal microbiota is increasingly recognized as an important component of host health, metabolism and immunity. Early gut colonizers are pivotal in the establishment of microbial community structures affecting the health and growth performance of chickens. White Lohmann layer is a common commercial breed. Therefore, this breed was selected to study the pattern of changes of microbiota with age. In this study, the duodenum, caecum and colorectum contents of white Lohmann layer chickens from same environment control farm were collected and analyzed using 16S rRNA sequencing to explore the spatial and temporal variations in intestinal microbiota. The results showed that the diversity of the microbial community structure in the duodenum, caecum and colorectum increased with age and tended to be stable when the layer chickens reached 50 days of age and the distinct succession patterns of the intestinal microbiota between the duodenum and large intestine (caecum and colorectum). On day 0, the diversity of microbes in the duodenum was higher than that in the caecum and colorectum, but the compositions of intestinal microbes were relatively similar, with facultative anaerobic *Proteobacteria* as the main microbes. However, the relative abundance of facultative anaerobic bacteria (*Escherichia*) gradually decreased and was replaced by anaerobic bacteria (*Bacteroides* and *Ruminococcaceae*). By day 50, the structure of intestinal microbes had gradually become stable, and *Lactobacillus* was the dominant bacteria in the duodenum (41.1%). The compositions of dominant microbes in the caecum and colorectum were more complex, but there were certain similarities. *Bacteroides*, *Odoribacter* and *Clostridiales vadin BB60 group* were dominant. The results of this study provide evidence that time and spatial factors are important factors affecting the intestinal microbiota composition. This study provides new knowledge of the intestinal microbiota colonization pattern of layer chickens in early life to improve the intestinal health of layer chickens.

## 1. Introduction

It is well-known that the layer chicken gut microbiota influences the host, gut development and plays important regulatory role in host immunity and overall health [1,2]. With the development of sequencing technology, research on gut microbiota interactions with their host has become deeper in recent years [3,4]. Some researchers have shown that the gut microbiome plays an instrumental role in the development of the host immune system by promoting immune system development, immune homeostasis and limiting pathogen colonization [5]. The stable community was destroyed, which not only led to host metabolic disturbance but also had great and even irreversible impacts on the immune system. In addition, gut microbes can help the host breakdown some nondigestible foods, such as cellulose, produce bacterial proteins or synthesize vitamins for the body to digest and absorb well.

It is now widely recognized that the early life is a critical window for the acquisition and colonization of the host gut microbiome [6,7]. The acquisition and establishment of the gut microbiota early in life is crucial for chickens, since early gut colonizers are pivotal in the establishment of permanent microbial community structures affecting the health and growth performance of chickens [8]. The microflora first colonizing in the intestinal tract can interfere with the later colonization of microorganisms in the intestinal tract by preempting or changing the niche [9]. Ballou (2015) found microbes in the gut of broilers at day 1 are mainly *Enterobacteriaceae* [10] microbes that can be acquired before it hatches directly from the oviduct of the hen [11] or from the environment through the pores in the eggshell [12]. The intestinal tract of newly hatched chicks starting to mature through a series of morphological, biochemical and molecular structure changes occurred in the following two weeks [13]. Once the chicks arrive on the farm, they will be exposed to a more diverse microbial environment. Microbes are ingested from feed, water and air. Succession occurs rapidly, and the differentiation of populations in different intestinal segments occurs in the early life [14]. After the chicks began to feed, *Lactobacillus* was initially colonized in the crop and small intestine and were established within two weeks and became the dominant bacteria [15]. The caecum microbiota evolved to be dominated by facultative anaerobic bacteria, including *Bacteroides*, *Peptostreptococcus* and *Eubacterium* [16]. It has been reported that the microbial community structure tends to be stable when the chickens reach around 40 days [15,17].

There are distinct different functions along the gut of chickens and the microenvironment changes with different functions [18]. The intestinal tract of chickens is composed of a small intestine and large intestine. The small intestine ecosystem is principally relevant to digestive health, because the duodenum and jejunum are primarily responsible for nutrient digestion and absorption in animals [19]. After the food is initially digested in the stomach, the resulting chyme enters the small intestine [20]. There is significant contact between chyme and commensal bacteria in the small intestine [21]. These bacteria are involved in degrading carbohydrates, proteins and fat into small molecules that can be easily absorbed through the epithelium of the small intestine. *Lactobacillus* can produce a variety of enzymes to promote the absorption of nutrients and produce acidic compounds [22], reduce pH, increase calcium absorption and affect the colonization and reproduction of other acid nonresistant bacteria. *Bacillus subtilis*, on the other hand, can promote digestion and absorption of proteins and improve the digestibility of nutrients by secreting protease and amylase [23]. The cecal microbiota has been suggested to play an important role in nutrition via the production of short-chain fatty acids, nitrogen recycling and amino acid production [24]. *Bifidobacterium* can inhibit the growth of intestinal pathogenic bacteria and the formation of toxic metabolites by producing acetic acid and lactic acid, stimulate intestinal peristalsis and reduce the intestinal overabsorption of water [25], while *Coprococcus* can ferment carbohydrates and use formic acid and propionic acid to produce butyric acid and acetic acid [26]. These beneficial bacteria can produce a variety of short-chain fatty acids (SCFA), nourish intestinal epithelial cells, promote intestinal absorption of various nutrients and inhibit the reproduction of pathogenic bacteria by reducing intestinal pH [27,28,29]. In addition, the cecal microorganisms play an important role in uric acid catabolism [30]. Three to six hours after hatching, the cecal bacteria can decompose uric acid under anaerobic conditions. *Peptostreptococcus* and *Clostridium* have the ability to decompose uric acid [31].

Therefore, an insight into how the microbiota varies over time in different segments will help better understand the microbial communities of layer chickens. In this study, we investigated the diversity and community variations in the intestines of layer chickens from hatching to 57 days by high-throughput 16S rRNA gene sequencing. Additionally, we constructed a microbial development model of different intestinal segments of layer chickens, which was used to study the maturation of the intestinal microbiota of layer chickens in the natural state. Our results not only suggest shifts and differences in gut microbial communities in layer chickens over time but also provide a theoretical basis for the accurate regulation of intestinal microorganisms in layer chickens.

## 2. Materials and Methods

### 2.1. Bird Management and Sample Collection

Samples for this study were collected on site from a commercial enclosed cage layer chicken farm in Guangdong Province, China from June to August 2019. Lohmann was used as single-breed animal model as applied by Videnska (2014) for this study [32]. All layer chickens were kept within the same house under similar conditions throughout the entire study. Each house was 80-m-long, 15-m-wide and 4-m-high, raising approximately 14,000 layer chickens. The manure produced in the layer chicken house was removed daily by mechanical scrapers. The house was installed with water curtains at the front end and 8 fans at the rear end for negative pressure ventilation during the summer.

The layer chickens were given ad libitum access to feed (Table 1) and received water freely throughout the day. Six birds were randomly obtained from the flock in the middle of house at days 0, 1, 3, 5, 7, 12, 18, 24, 30, 36, 40, 42, 50 and 57 after hatching, with a total of 84 layer chickens used for this study. The above sampling time points were selected based on the fact that the maturation time of chicken intestinal microbes is approximately 40 days [17,33]. The six birds selected for each time point were killed by cervical dislocation. After the abdomen was opened; the duodenum, caecum and colorectum were removed from each chick, and the different intestinal segments were immediately placed in liquid nitrogen and collected, transferred to the laboratory and stored at −80 °C until DNA extraction (Appendix A
Appendix A).

### 2.2. DNA Extraction, Library Preparation and Sequencing

Total genomic DNA was extracted from the duodenum, caecum and colorectum using a commercially available QIAamp PowerFecal DNA Kit (Qiagen, Hilden, Germany). Briefly, 750 μL of PowerBead and 60 μL of C1 were added to the 200-mg sample, vortexed at maximum speed for 10 min to homogenize and incubated at 70 °C for 5 min. Then, samples were centrifuged at 13,000× *g* for 2 min, and 600 μL of supernatant was treated as recommended by the manufacturer. In the process, we set up a negative control to check whether artefact sequence contamination existed in this study. DNA concentration was assessed using a Qubit 2.0 fluorometer (Invitrogen, Thermo Fisher Scientific, Waltham, MA, USA), and the integrity was checked by 1% agarose gel electrophoresis. Extracted DNA was used as a template for PCR using barcoded primers to amplify the V4 region of the 16S rRNA gene. The V4 region of the 16S rRNA gene was amplified using universal primers, and PCR was carried out in triplicate using 10-μM primers 515F (5’-GTGYCAGCMGCCGCGGTAA-3’) and 806R (5’-GGACTACNVGGGTWTCTAAT-3’) [34], 1× GoTaq Green Master Mix (Promega, Madison, WI, USA), 1-mM MgCl_2_ and 3 μL of DNA template or nuclease-free water as a negative control. The amplification conditions included an initial denaturation of 3 min at 94 °C, followed by 35 cycles of 94 °C for 45 s, 50 °C for 60 s and 72 °C for 90 s and a final extension for 10 min at 72 °C. PCR products were pooled at equimolar concentrations and purified using a QIAquick PCR purification kit, and 250-bp read sequencing was performed on the Illumina HiSeq platform. Raw sequencing data in this study were deposited in the European Nucleotide Archive database with the accession number PRJNA672781. In addition, 3-, 12-, 30-, 40- and 50-day samples were selected for qPCR. The primer sequences for 16S rRNA were 5′-CTGGAACTGAGACACGGTCC-3′ (forward) and 5′-GGTGCTTCTTCTGCGGGTAA-3′ (reverse). The absolute abundance of each sample was calculated as follows: absolute abundance (copies/mL) = labelling template abundance (copies/μL) × sample DNA elution volume (μL)/sample weight (mg). The qPCR results of the total bacteria showed that the copy number of bacteria in the negative control group was tiny (0.21 ± 0.15 log10 (copies/mL)) and significantly lower (*p* < 0.01) than that in the duodenum, caecum and colorectum samples. Therefore, the negative control group was not considered in the subsequent analysis (Appendix A
Appendix A).

### 2.3. Bioinformatics and Statistical Analysis

The raw reads of 16S rRNA gene sequencing were demultiplexed and quality-filtered using the Quantitative Insights into Microbial Ecology QIIME2 platform. Reads were trimmed and removed based on quality scores < 25 and lengths > 225 bp, respectively [35]. Sequence denoising was performed to obtain amplicon sequence variants (ASVs) for species classification with similarity [36]. The ASVs abundance data were normalized using a standard sequence number corresponding to the sample with the fewest sequences. The Chao1 and Shannon indices were also determined in QIIME2. Principal coordinates analysis (PCoA) was performed on Bray-Curtis, unweighted UniFrac distance metrics to visualize the relationships among the samples. PICRUSt was used to predict metagenome function by 16S rRNA marker gene sequences and referencing published complete genome sequences [37]. Linear discriminant analysis (LDA) effect size (LEfSe) was used to identify stage-dependent features and functional genes showing differential abundance between groups (LDA score > 2.0). The R package of “corroplot” was used to generate the heat maps. The R package “psych” was used to calculate the Spearman’s correlation coefficient. Cytoscape 3.30 was used for network building. Cooccurrence patterns of genera were constructed in the network interface by Spearman’s rank correlations based on bacterial abundance. A valid cooccurrence event was based on strong (Spearman’s *r_s_* < −0.5 or *r_s_* > 0.5) and significant (*p* < 0.05) correlations between genera.

### 2.4. Modelling the Maturation Process of the Gut Microbiota Using the Random Forest Algorithm

Random forest models were used to regress the relative abundances of ASVs in the time-series profile of the microbiota of layer chickens against their chronologic age, using default parameters of the R implementation of the algorithm (R package “randomForest”, ntree = 10,000, using default mtry of p/3, where p is the number input 97% identity ASVs (features)) [33]. The random forest machine-learning algorithm was used to determine a ranked list of all bacterial taxa in the order of age-discriminatory importance. The “rfcv” function was applied over 100 iterations to estimate the minimal number of top-ranking age-discriminatory taxa required for a prediction. A sparse model with 30 predictors (variance explained) was selected on the basis of 10-fold cross-validation. A smoothing spline function was fit between microbiota age and chronologic age of the layer chickens in the validation sets to which the sparse model was applied.

## 3. Results

### 3.1. Similarity of the Microbial Community Structure between Sample Types

In this study, after quality filtering and assembly, we obtained 252 high-quality samples that were used for the downstream analysis, including samples of the duodenal contents (*n* = 84), cecal contents (*n* = 84) and colorectal contents (*n* = 84) (Appendix A
Appendix A). The total high-quality sequences were 12,174,399, with an average of 48,311 sequences per sample (ranging from 19,234 to 119,031). The overall number of unique ASVs detected by the analysis reached 5085. The rarefaction curves based on the observed and Shannon indexes of all samples nearly reached a plateau, indicating that the sampling depth was sufficient to characterize the bacterial communities (Appendix A
Appendix A).

The alpha diversity analysis revealed that the species richness and diversity of the microbial communities were distinct for different sources. We observed higher Shannon diversity indexes in the duodenal microbiome at the first time point (day = 0), and then, they decreased shortly (to the bottom at day 7 for the Chao1 and Shannon diversity indexes) and rebounded over time. The caecum and colorectum results were different from the duodenum results. The overall alpha diversity increased over time, starting from the first time point (day = 0), as demonstrated by the Shannon and Chao1 indices (Figure 1).

To reveal the shifts in community membership and structure in the different intestinal segments of layer chickens of different ages, a principal coordinates analysis (PCoA) was performed based on Bray-Curtis distances. Intestinal samples of the duodenum, caecum and colorectum all showed that the distances between day 0 and day 1 were short. These results indicated that the intestinal microbiome community was similar within 48 h after newly hatched chickens. However, they became progressively divergent with age (Figure 2b–d). The microbial succession patterns in the caecum and colorectum were similar but different from those in the duodenum (Figure 2a). These observations suggested that time and intestinal segment are important factors affecting the intestinal microbiota composition and that the intestinal microbiome community changed significantly over time.

By comparing 16S rRNA sequences with the Greengene database, at the phylum level, a total of 21 phyla in the duodenum, 17 phyla in the caecum and 16 phyla in the colorectum were identified. *Firmicutes*, *Proteobacteria* and *Bacteroidetes* were the three most abundant phyla in all the samples. In the duodenum, they accounted for 13–87%, 2.6–50.2% and 1.8–13%, respectively. In the caecum, they accounted for 31.9–80.0%, 1.8–57% and 0.1–37%, respectively. They accounted for 26.6–90.5%, 1.3–70.4% and 1.2–29.5% of the total sequence in the colorectum, respectively. The relative abundance of *Firmicutes* and *Bacteroides* increased over time, while the relative abundance of *Proteobacteria* decreased over time (Figure 3).

At the genus level, a total of 325 genera in the duodenum, 231 genera in the caecum and 239 genera in the colorectum were identified, and the 15 most abundant microbiome features at the genus level were selected in stacked bar charts (Figure 4). Most of the dominant genera in the intestine of layer chickens belonged to *Firmicutes* and *Bacteroides*. We observed that, on day 0, the compositions of the different intestinal microbes were relatively similar, with *Escherichia* and *Clostridium* as the main microbes (duodenum: 46.4% and 2.2%, caecum: 56.4% and 33.3% and colorectum: 60.5% and 17.2%). However, the intestinal microbiome gradually increased in richness and diversity with age. On day 3, a large number of *Lactobacillus* began to colonize the different segments of the intestines (46.3%) and reached a maximum on day 18 (75.5%), after declining slightly. By the end of experiment, the relative content of *Lactobacillus* in the duodenum was 38.7%. These results indicated that the duodenum was the main colonization segment of *Lactobacillus*. In addition, in the caecum and colorectum, the anaerobic bacteria *Lachnospiraceae* and *Ruminococcaceae* increased over time. In the caecum, they accounted for 20.1% and 6.5%, respectively. They accounted for 12.3% and 5.4% in the colorectum, respectively. The microbiome structure was similar between the caecum and colorectum. On day 50, *Bacteroides* (14.7% in the caecum and 9.7% in the colorectum), *Odouribacter* (14.3% in the caecum and 9% in the colorectum) and *Clostridiales vadin BB60 group* (8.2% in the caecum and 6.8% in the colorectum) were dominant, while the relative abundance of *Escherichia* was relatively low. As expected, the duodenum, cecum and colorectum microbiota experienced a dramatic shift with age.

### 3.2. Maturation of the Layer Chickens Intestinal Microbiota

Stage-associated bacterial features were identified by using LEfSe [38], an algorithm that focuses not only on statistical significance but, also, on the biological consistency. The stage-associated bacterial abundances of these features were visualized using a heat map (Figure 5a–c). The LEfSe analysis confirmed most of the observations mentioned above (Appendix A
Appendix A). For example, the relative abundances of *Clostridium* and *Escherichia* nitrogen-fixing microbes before day 1 were higher than those for other days; in addition, *Oscillibacter* (urease-producing bacteria) had a higher abundance in the caecum and colorectum at 12 days than at other times. However, *Lactobacillus* had higher abundances in different intestinal segments after 18 days. *Bacteroides* had a higher abundance in the caecum at 30 days than at other times. Overall, the duodenum and colorectum had more diverse bacteria at 12 days, while the caecum had more diverse bacteria at 18 days.

To probe the age-dependent development of microbiota in the intestines of layer chickens, the relative abundances of the amplicon sequence variants (ASVs) were regressed against the chronologic age of each chicken using a random forest machine-learning algorithm. The top-ranking age-discriminatory taxa were selected according to their variable importance measures using 10-fold cross-validation. Thus, the top 30 age-discriminatory taxa were identified and used for the subsequent construction of a microbiota-based model for discriminating the degree of microbiota maturity, as the inclusion of any taxa beyond these top taxa produced only minimal improvements in the model performance. This model, which includes 13 genera, was able to distinguish the maturity of the intestinal microbiota over 57 days (Appendix A
Appendix A). The natural development of the microbiota exhibited a smooth curve and gradually changed until reaching a plateau at day 50 (Figure 5d–f). This indicates that the gut microbiota had reached maturity by the end of this study and that the intestinal microbial composition was relatively stable. These age-discriminatory taxa were primarily affiliated with *Lactobacillus* and *Bacteroides*, and the relative abundances of these age-discriminatory taxa significantly changed across the sampling times.

To explore bacterial interactions within the layer chicken intestine samples, we used a network analysis based on strong (Spearman’s *r_s_* < −0.5 or *r_s_* > 0.5) and significant (*p* < 0.05) correlations between the bacterial genera, with an average relative abundance greater than 0.5% in intestinal tracts of the layer chickens (Figure 6a–c). In this network, it was assumed that these microbes interacted with each other in either a positive or negative manner. The duodenum network consisted of 15 nodes (genera) and 58 edges (relation). The caecum network consisted of 30 nodes (genera) and 478 edges (relation). The colorectum network consisted of 31 nodes (genera) and 297 edges (relation). In the duodenum, most genera showed as positively correlated with others (*Alistipes*, *Bacteroides* and *Odoribacter* were a positive correlation (*p* < 0.05)), and only *Pediococcus* and *Helicobacter* were significantly negative (*p* < 0.05). Microbial interactions in the cecum and colorectum were more complex. In the caecum, there were 297 positive and 181 negative correlations among the 30 genera. Among them, *Clostridioides* interacted with 22 other genera. It was negatively correlated with *Mollicutes RF39*, *Campylobacter* and *Barnesiella* and positively correlated with *Enterobacteriaceae*, *Clostridium* and *Escherichia*. *Bacteroides* was positively correlated with *Barnesiella*, *Odoribacter* and *Gastranaerophilales* and negatively correlated with *Enterobacteriaceae*, *Clostridioides* and *Caproiciproducens*. In the colorectum, *Clostridium* was negatively correlated with *Clostridiales vadinBB60*, *Gastranaerophilales* and *Alistipes* and positively correlated with *Enterobacteriaceae*, *Clostridioides*, *Enterococcus* and *Escherichia*. *Barnesiella* interacted positively with *Bacteroides*, *Gastranaerophilales* and *Faecalibacterium* and negatively with *Enterococcus*, *Clostridioides* and *Enterobacteriaceae*.

### 3.3. Potential Functional Profile of the Gut Microbial Community

To investigate the changes in the intestinal microbiota functional profiles of the duodenum, caecum and colorectum samples, KEGG Orthology (KO), KEGG Enzyme (EC) and pathways were predicted by comparing the PICRUSt software results with the MetaCyc database. We identified 5984 KOs in the duodenum, 5505 KOs in the caecum and 5698 KOs in the colorectum; 1861 ECs in the duodenum, 1692 ECs in the caecum and 1752 ECs in the colorectum and 372 pathways in the duodenum, 341 pathways in the caecum and 348 pathways in the colorectum. To identify different enrichments of the functional capacities in the duodenum, caecum and colorectum samples, we performed a LEfSe analysis using the relative abundances of the KOs (Appendix A
Appendix A), ECs (Figure 7a,b) and pathways (Appendix A
Appendix A). Through the LEfSe analysis, it was found that only one KO (K07258, dacC, dacA and dacD; serine-type D-Ala-D-Ala carboxypeptidase) enriched in the duodenum at day 12. Under the duodenal acidic environment, serine-type D-Ala-D-Ala carboxypeptidase has the activity of proteolytic enzymes and can participate in the synthesis, modification and degradation of peptides and proteins [39,40,41,42,43,44,45,46]. The ECs were related to the nitrogen metabolism (glutaminase and pyridoxal 5’-phosphate synthase) in the caecum and colorectum at day 1. In addition, the EC related to energy (nucleotidase) was more abundant in the colorectum at day 1 and related to monosaccharides (phosphofructokinase) at day 12.

## 4. Discussion

The intestines of chickens are colonized with complex microbial communities, which are known to play an important role in the overall health and performance of the chickens [43,47,48,49,50,51]. Since the intestinal microbiota varies greatly along the intestinal tract and undergoes substantial changes over time [44], it is important to understand initial spatial and temporal variation of microbiota in different segments of the layer chickens. Commercially produced chicken gut microbiota development is characterized by an absence of contact between newly hatched chicks and adult hens; therefore, the initial colonization of the newly hatched chick gut may be dependent on the microbiota present in the hatchery or the housing environment [32]. When hatched chicks are delivered from the hatchery to rearing chick houses, their intestinal microbiota is very simple and contains only a very small number of bacteria belonging to a few species [45,46]. These bacteria derived from the environment can gain entry into the immature intestine and then colonize [16]. The results showed that the diversity and abundance of intestinal microbiota were low after hatching. The intestinal microbiota underwent a series of successions and became diverse and rich over time. Among them, the abundance and diversity of microbiota in the caecum and colorectum increased, and the microbial diversity in the duodenum increased, but the abundance was always low. Interestingly, we observed distinct succession patterns of the intestinal microbiota between the duodenum and large intestine (caecum and colorectum). Despite a similar initial composition after hatching, the intestine underwent progressive expansion and diversification as soon as day 3. These results suggest that each segment of the intestine developed its own unique microbial community as the layer chickens matured. It is obvious that time and spatial factors are important factors affecting the intestinal microbiota composition [49].

While the succession of microbiota communities was different in each segment of the intestine, we can broadly say that the microbiota communities were predominantly initially formed by a low diversity community. The results show that Proteobacteria were most abundant in all segments of the intestine at day 0. There was approximately 50.2% in the duodenum, 57.0% in the caecum and 62.0% in the colorectum. However, the phylum Firmicutes became the dominant phylum as the layer chickens matured. There was approximately 87.8% in the duodenum, 59.5% in the caecum and 58.4% in the colorectum at day 18. For laying hens, it was shown that facultative anaerobic Proteobacteria are the most abundant phylum in early life [10,32]. The main reason is that the gut of newborn animals still contains some oxygen, so facultative anaerobic proteobacteria have priority for rapid colonization in the gut [50]. Although these “pioneer” microbes are gradually replaced by obligate anaerobes after being selected by the intestinal niche, facultative anaerobes can create an anaerobic environment for obligate anaerobes that efficiently utilize complex polysaccharides by scavenging oxygen [51,52]. This also explains why facultative anaerobes maintain a certain abundance in the intestinal tract of layer chickens.

The microbial composition was different in each section of the gut [53]. Our study found that *Escherichia* were the common dominant bacteria of different intestinal segments for the first 48 h. Between day 3 and day 7, the proportion of *Escherichia* greatly decreased in all of our sample types, and while it remained present at later timepoints, it was never highly abundant. This is possibly caused by the *Escherichia* involved in scavenging oxygen to contribute to an anaerobic environment [52]. These activities may explain the dominant level of *Escherichia* in early life. The duodenum has a low pH that limits the growth of bacteria and most pathogens [54]. At seven days, the duodenum was mainly composed of *Lactobacillus*, *Lachnospiraceae* and *Enterobacteriaceae*. Among them, *Lactobacillus* is the most important genus and has a high relative abundance for a long time. Compared with pathogenic bacteria, *Lactobacillus* has a strong adhesion to the intestinal wall of animals, competitively inhibits the adhesion of pathogenic bacteria, effectively inhibits the reproduction of pathogenic bacteria and maintains the balance of intestinal flora [55,56]. In addition, the small intestine microbiota could positively contribute to the nutrient digestion and absorption processes [57]. On day 12, the duodenum was enriched with proteolytic Carboxypeptidases. The caecum and colorectum microbiota are richer and more diverse, mainly in anaerobes [58]. Chickens release the caecum contents into its colorectum several times per day, which may result in the higher similarity between the microbial communities of the colorectum and cecum in birds [59]. On day 18, the caecum and colorectum were mainly composed of *Bacteroides*, *Ruminococcaceae* and *Clostridiales vadinBB60 group*. On day 50, the caecum and colorectum were mainly composed of *Bacteroides*, *Ruminococcaceae* and *Alistipes*.

In addition, our study explored the gut microbiota maturity of newly hatched layer chickens using a random forest regression model, and the results indicated that the intestinal microbiota of layer chickens reached maturation at day 50. However, several studies have suggested that feeding prebiotics and dietary fibers can improve gut health and shorten the maturation time of chickens. For example, Gao (2017) reported that the maturation time of broiler intestinal microbes is approximately 30 days, and the supplementary feeding of *Lactobacillus plantarum strain 8* accelerated the maturation of broiler intestinal microbiota to 15 days by promoting the growth of indigenous *Lactobacillus* [33]. It is possible that probiotics such as *Lactobacillus* produce lactic acid and short-chain fatty acids in chicken intestines, which reduces the intestinal pH value. The resulting more acidic environment prevents the growth of other intestinal bacteria, such as *Salmonella*, *Campylobacter* and *Clostridium*, but promotes the growth and higher diversity of indigenous *Lactobacillus* [60,61]. Therefore, we speculate that supplementing probiotics to the diet of layer chickens in early life can promote intestinal maturation. However, whether this has any effect on the health and production performance of layer chickens needs further research.

## 5. Conclusions

Most studies of layer chicken intestinal microbiotas have focused on the succession in one segment over time; few studies have compared the variations in multiple segments from the same set of layer chickens. At hatching, *Escherichia* occupied the colonization sites faster than other bacteria and is the main microbe in the intestines. Over time, the abundances of the microbial structure changed, and when the intestinal microbes reached maturity at day 50, *Lactobacillus* was the dominant microbial genus in the duodenum, and *Bacteroides*, *Odoribacter*, and *Clostridiales vadin BB60 group* were the dominant microbes in the caecum and colorectum, while the abundance of *Escherichia* was relatively low. This study provides insight into the spatial and temporal variations in the intestines of layer chickens, which may provide new ideas and directions for the optimal time to regulate intestinal microbes to improve intestinal health.

## Figures and Tables

**Figure 1 animals-11-00840-f001:**
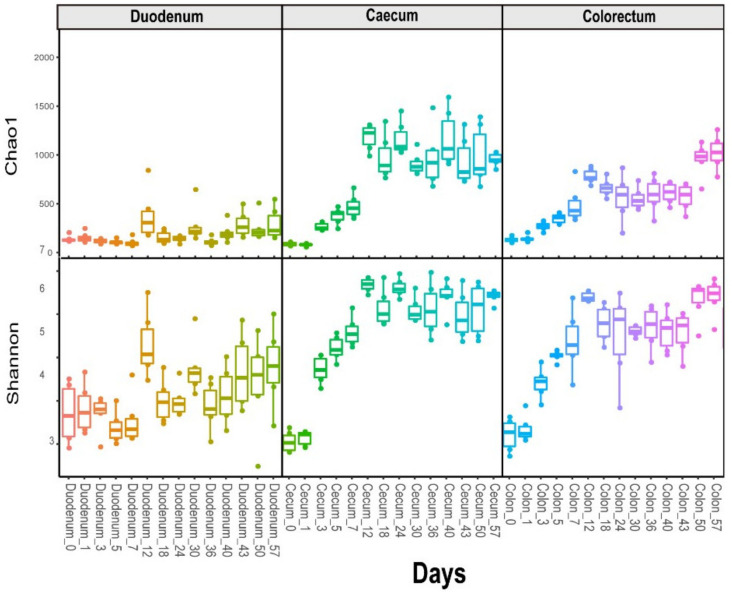
Longitudinal exploration of the microbiome in layer chickens. The alpha diversity indices (Chao1 and Shannon) with age across layer chickens in the duodenum, caecum and colorectum.

**Figure 2 animals-11-00840-f002:**
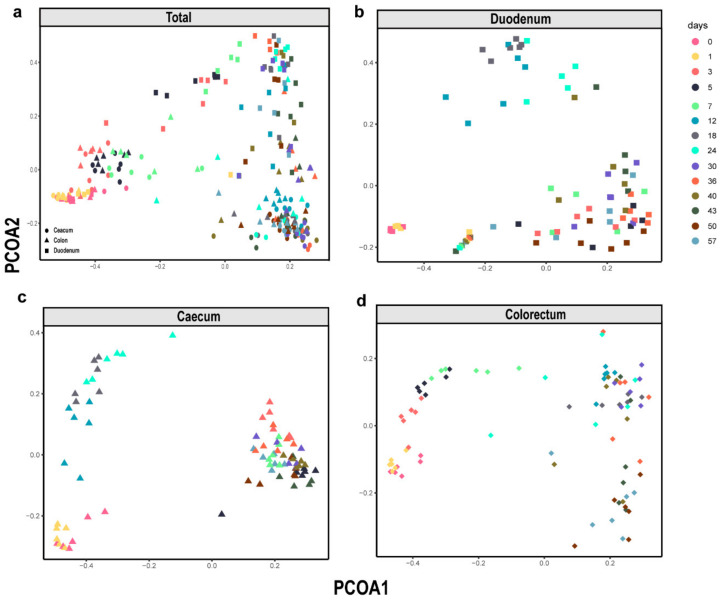
β-Diversity of the microbiota in all types of samples. Principal coordinates analysis (PCoA) plot of the Bray-Curtis dissimilarity of the bacterial communities from the layer chicken. (**a**) Total sample, (**b**) Duodenum, (**c**) Caecum and (**d**) Colorectum. Each point represents a different sample calculated using Bray-Curtis distance according to the amplicon sequence variant (ASV) compositions and abundances.

**Figure 3 animals-11-00840-f003:**
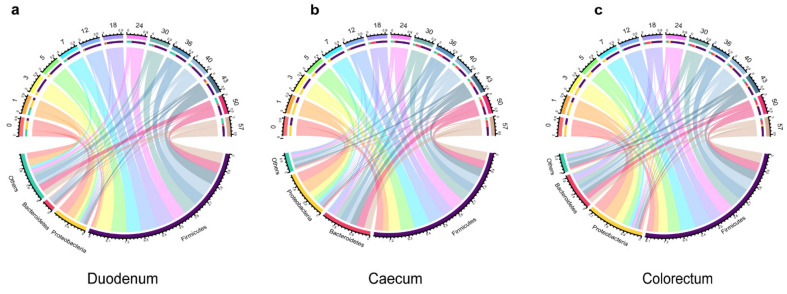
Relative abundance of the phylum in different types of samples. The upper part of the circle represents different days; the lower part indicates the relative abundance of microorganisms at the phylum level and the area size indicates the relative abundance, duodenum (**a**), caecum (**b**) and colorectum (**c**) of layer chickens.

**Figure 4 animals-11-00840-f004:**
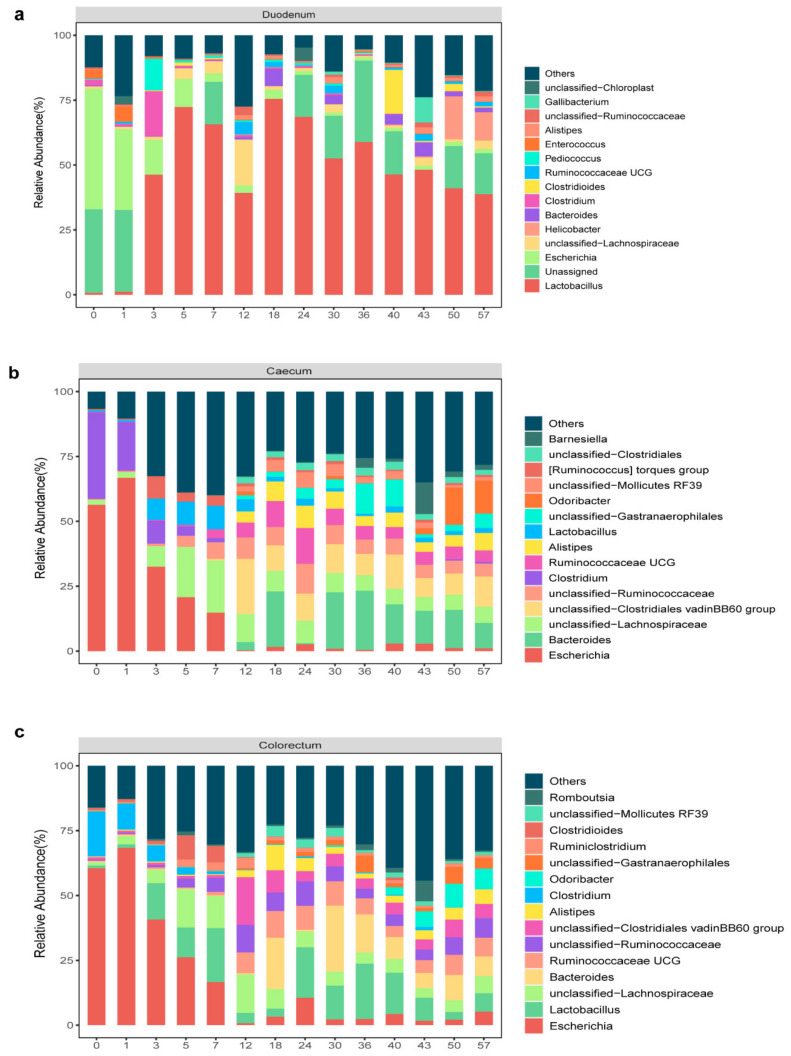
Relative abundance of the genus in different types of samples. Stacked bar chart displaying the change in the average relative abundance of the top 15 genera in the duodenum (**a**), caecum (**b**) and colorectum (**c**) of layer chickens.

**Figure 5 animals-11-00840-f005:**
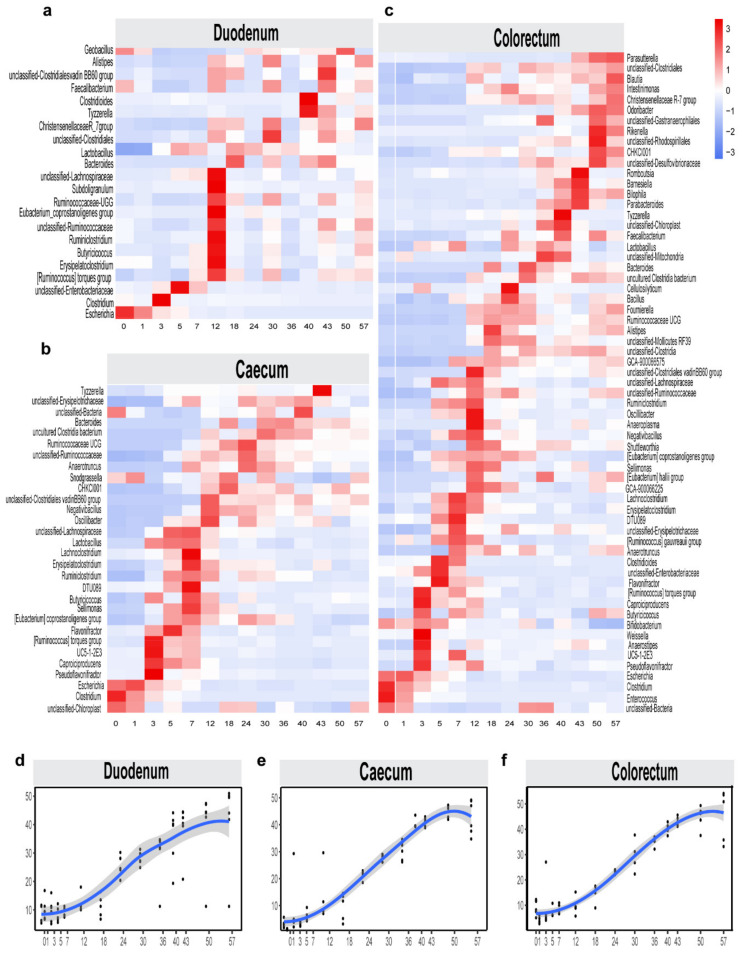
Bacterial taxonomic biomarkers for defining different intestinal microbiota maturation in layer chickens throughout the first 57 days of life. Heat map displaying the average relative abundance of stage-associated microbiota in duodenum (**a**), caecum (**b**) and colorectum (**c**) identified by LEfSe (linear discriminant analysis (LDA) > 2) in the test trial. The maturity index for the duodenum, caecum and colorectum at a given time point was calculated through the random forest model: the predicted time point was used as the *Y*-axis coordinate to represent the degree of maturity, while the actual time point was used as the *X*-axis coordinate. The temporal patterns of intestinal maturation were compared among the duodenum (**d**), the caecum (**e**) and the colorectum (**f**), which indicated intestinal microbiota maturation at 50 days.

**Figure 6 animals-11-00840-f006:**
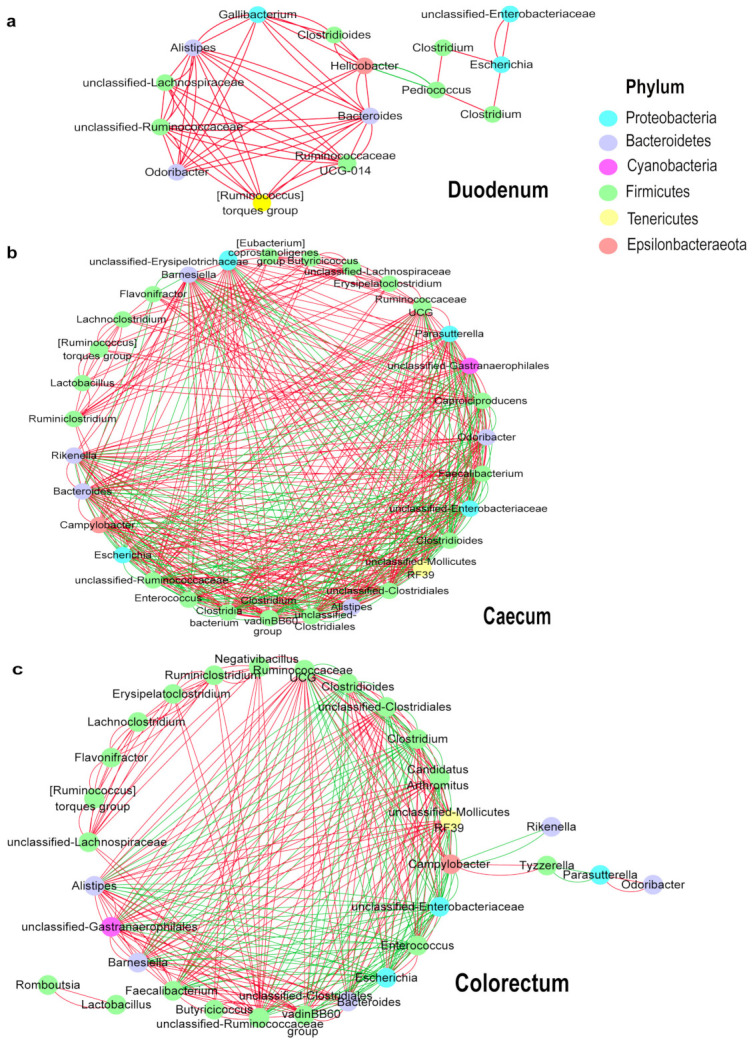
Network of co-occurring genera within layer chicken intestines. The important correlative associations between genera were determined based on the Spearman’s algorithm. The nodes represent the genera, and the edges represent strong and significantly positive (**red**) or negative (**green**) correlations between genera. (**a**): Duodenum, (**b**): Caecum, (**c**): Colorectum.

**Figure 7 animals-11-00840-f007:**
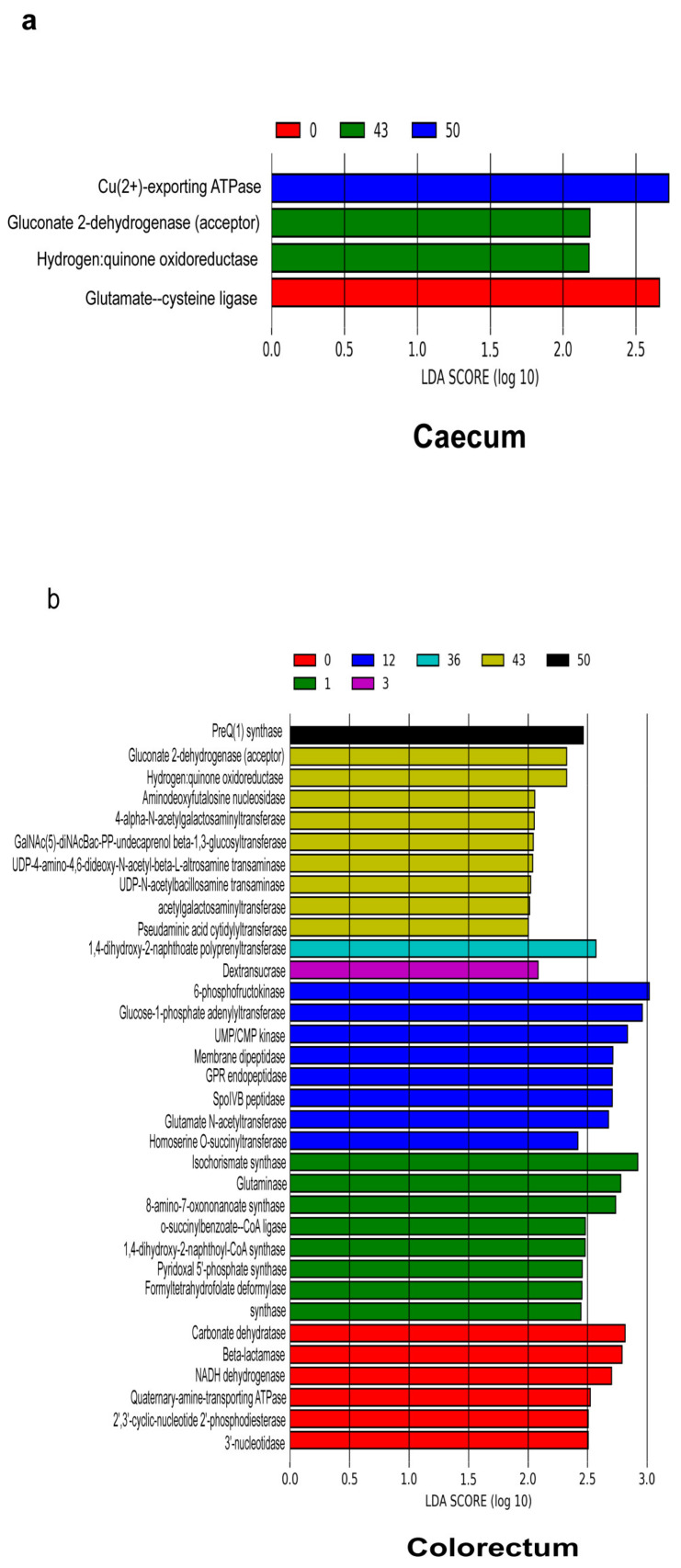
Functional predictions for the microbiota in different samples. The result of the LEfSe analysis is based on the PICRUSt dataset, which was conducted to identify EC that differentiated functional enzymes between the caecum (**a**) and colorectum (**b**) at different time points. Modules with a linear discriminant analysis (LDA) score > 2.0 are plotted.

**Table 1 animals-11-00840-t001:** Ingredient and nutrient compositions of the diet.

Ingredients	Content (%)	Nutrient Levels	Content
Maize meal	52.48	ME (MJ/kg)	27.50
Soybean meal	31.83	crude protein (%)	18.50
Cottonseed meal	5.99	Ca (%)	1.00
Rapeseed meal	6.84	TP (%)	0.70
Calcium carbonate	0.76	AP (%)	0.45
Calcium hydrogen carbonate	1.15	Na (%)	0.16
Salt	0.35	Met (%)	0.38
Trace elements	0.43	Lys (%)	1.00
Compound microorganism	0.17		
Total	100		

## Data Availability

The raw sequencing data in this study was deposited in the European Nucleotide Archive database with the accession number PRJNA672781.

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
