# Peer review of "Microbial Diversity and Community Variation in the Intestines of Layer Chickens"

_animals, 2021, doi:10.3390/ani11030840_

Round 1
Reviewer 1 Report
The article titled Microbial diversity and community variation in the intestines of layer chickens by Xioa et. al. mainly focuses on the variation of gut microbiota in different sections of the gut of layer chickens with age. Although the introduction of the article gives appropriate understanding, yet it needs to be reorganized and improved. Besides this, there are several other comments that authors need to address.
- Line 19: Please replace the word “early” with the word “different”.
- Line 25 and 26: Please replace “This study provides a reference for improving the intestinal health of layer chickens by regulating intestinal microbes” with “This study provides information about changes in microbiota composition of layer hens with age.”
- Line 29: Please remove the word “permanent”.
- Line 30 and 31: Please rewrite to improve understanding. I assume the authors mean that White Lohmann layer is a common commercial breed. Therefore, this breed was selected to study the pattern of changes of microbiota with age.
- Line 40: Please provide names of bacteria.
- Line 49: Please replace the word “Intestine” with “intestinal”.
- Line 50: Please remove the words “early stage”.
- Line 53: Please remove “,”.
- Line 54: Please replace the word “physiology” with immunity and add the word “overall” before the word “health”.
- Line 66: Please remove the word “in”.
- Lines 72-76: Please include studies relevant to chicken only.
- Line 79: Please replace the words “was immature, and” with the words “started to mature through”.
- Line 82: Please replace the word “intestine” with “intestinal” and remove the word “at”.
- Line 86-88: Please rewrite to improve the understanding.
- Line 129: Please remove the word “old”.
- Line 131: Please remove “(“.
- Line 134: Please remove the word “level”.
- Line 237 and 238: Please rewrite to improve the understanding.
- Line 260: Please replace the word “intestine” with “intestinal”.
- Line 271: Please insert the word "of" between the words “development” and “microbiota”.
- Lines 285-289: Please mention the names of all genera that are positively correlated and negatively correlated.
- Line 326: Please clarify the statement “It was related to hydrolyzed protein”.
- Lines 339-343 and 104-105: Please avoid replication.
- Line 370: Please clarify the words “incomplete closed gut”.
- Lines 385-387: Please explain and clarify these lines
- Line 389: Please rewrite to improve the understanding.
- Lines 399-405: This study mainly focuses on the variation of gut microbiota of layer chickens with age but here the authors are taking about feeding probiotics and feeding methods. Please elaborate the connection.
- Lines 414-416: Please rewrite. There are certain studies available on the analysis of gut microbiota of chickens. One such example is "Cui, Y., Wang, Q., Liu, S., Sun, R., Zhou, Y., & Li, Y. (2017). Age-Related Variations in Intestinal Microflora of Free-Range and Caged Hens. Frontiers in microbiology.
Author Response
Point 1: The article titled Microbial diversity and community variation in the intestines of layer chickens by Xioa et. al. mainly focuses on the variation of gut microbiota in different sections of the gut of layer chickens with age. Although the introduction of the article gives appropriate understanding, yet it needs to be reorganized and improved. Besides this, there are several other comments that authors need to address.
Response 1:Thank you, we have reorganized and improved the introduction according to your suggestion. Please refer to line 64-68,75-79,93-100 and line 109-130.
Point 2: Line 19: Please replace the word “early” with the word “different”.
Response 2: The word “early” has been replaced with “different” as suggested. Please refer to line 18.
Point 3: Line 25 and 26: Please replace “This study provides a reference for improving the intestinal health of layer chickens by regulating intestinal microbes” with “This study provides information about changes in microbiota composition of layer hens with age.”
Response 3: We have revised the sentence according to your suggestion from line 24 to line 25.
Point 4: Line 29: Please remove the word “permanent”.
Response 4: Removed as suggested. Please refer to line 29.
Point 5: Line 30 and 31: Please rewrite to improve understanding. I assume the authors mean that White Lohmann layer is a common commercial breed. Therefore, this breed was selected to study the pattern of changes of microbiota with age.
Response 5: Thank you and we have revised the sentence according to your suggestion. Please refer to lines 30-31.
Point 6: Line 40: Please provide names of bacteria.
Response 6: We have provided the name of bacteria according to your suggestion - lines 41-42.
Point 7: Line 49: Please replace the word “Intestine” with “intestinal”.
Response 7: We replaced the word “Intestine” with “Intestinal” as suggested. Please refer to line 50.
Point 8: Line 50: Please remove the words “early stage”.
Response 8: Removed as suggested. Please refer to lines 51.
Point 9: Line 53: Please remove “,”.
Response 9: Removed as suggested. Please refer to lines 55.
Point 10: Line 54: Please replace the word “physiology” with immunity and add the word “overall” before the word “health”.
Response 10: Thank you for the suggestion. We replaced the word “physiology” with “immunity” and add the word “overall” before the word “health” as suggested. Please refer to lines 55.
Point 11: Line 66: Please remove the word “in”.
Response 11: Removed as suggested. Please refer to lines 69.
Point 12: Lines 72-76: Please include studies relevant to chicken only.
Response 12: Removed studies not relevant to chicken as suggested. Please refer to lines 75-79.
Point 13: Line 79: Please replace the words “was immature, and” with the words “started to mature through”.
Response 13: Replaced the words “was immature, and” with the word “started to mature through” as suggested. Please refer to line 83.
Point 14: Line 82: Please replace the word “intestine” with “intestinal” and remove the word “at”.
Response 14: We have replaced the word “intestine” with the word “intestinal” and remove the word “at” as suggested. Please refer to line 87.
Point 15: Line 86-88: Please rewrite to improve the understanding.
Response 15: Thank you and we have revised the sentence according to your suggestion from lines 91-93.
Point 16: Line 129: Please remove the word “old”.
Response 16: The word “old” has been removed as suggested. Please refer to line 154.
Point 17: Line 131: Please remove “(“.
Response 17: Removed as suggested. Please refer to line156.
Point 18: Line 134: Please remove the word “level”.
Response 18: Remove the word “level” as suggested. Please refer to line 159.
Point 19: Line 237 and 238: Please rewrite to improve the understanding.
Response 19: Thank you and we have revised the sentence according to your suggestion from line 279 to line 280.
Point 20: Line 260: Please replace the word “intestine” with “intestinal”.
Response 20: Thank you for the suggestion. We replaced the word ““intestine” with the word “intestinal” as suggested. Please refer to line 301.
Point 21: Line 271: Please insert the word "of" between the words “development” and “microbiota”.
Response 21: Thank you for the suggestion. We have inserted the word "of" between the words “development” and “microbiota” as suggested. Please refer to line 314.
Point 22: Lines 285-289: Please mention the names of all genera that are positively correlated and negatively correlated.
Response 22: Thank you for the suggestion. We added the information of genera that are positively correlated and negatively correlated. Please refer to line 342 to line 355.
Point 23: Line 326: Please clarify the statement “It was related to hydrolyzed protein”.
Response 23: Thank you. We have clarified and rephrases the sentence according to your suggestion. Line 381 to line 383.
Point 24: Lines 339-343 and 104-105: Please avoid replication.
Response 24: Thank you and we have revised the sentence according to your suggestion from line 129 to line 130 and line 398 to line 403.
Point 25: Line 370: Please clarify the words “incomplete closed gut”.
Response 25: We have removed “incomplete closed” in the sentence. Line 431.
Point 26: Lines 385-387: Please explain and clarify these lines
Response 26: Thank you. We have clarified and rephrases the sentence according to your suggestion. Line 452 to line 453.
Point 27: Line 389: Please rewrite to improve the understanding.
Response 27: Thank you and we have revised the sentence according to your suggestion from line 457.
Point 28: Lines 399-405: This study mainly focuses on the variation of gut microbiota of layer chickens with age but here the authors are taking about feeding probiotics and feeding methods. Please elaborate the connection.
Response 28: Thank you and we elaborate the connection according to your suggestion from line 485 to line 492.
Point 29: Lines 414-416: Please rewrite. There are certain studies available on the analysis of gut microbiota of chickens. One such example is "Cui, Y., Wang, Q., Liu, S., Sun, R., Zhou, Y., & Li, Y. (2017). Age-Related Variations in Intestinal Microflora of Free-Range and Caged Hens. Frontiers in microbiology.
Response 29: Thank you. It was our mistake. We have revised the sentence according to your suggestion from line 505. We have added references to this literature, Ref No 53.

Reviewer 2 Report
The authors included all suggestions and corrections in the presented manuscript. All answers were replied adequate. They provided a detailed correction of the manuscript.
Author Response
Point: The authors included all suggestions and corrections in the presented manuscript. All answers were replied adequate. They provided a detailed correction of the manuscript.
Response: Thank you.
Reviewer 3 Report
Microbial diversity and community variation in the intestines of layer chickens
I firstly wanted to thank the authors for taking the time to alter and resubmit this article.
This revised and resubmitted article focusses on the microbial diversity found in the gut of layer chickens, in the first 57 days’ maturation. It was observed that the chicken gut microbiome stabilised around 50 days’ maturation, and there were distinct shifts in primary colonising species during this maturation period. The most prominent shift was from a primarily dominant lactobacillus colony, the microbiome adjusted to Bacteriodes, Odoribacter and Clostridales being more dominant. It is the authors hope that this work will be a reference point for further research on the microbiome of layer chickens.
Broad comments.
This article is well organised and attempts to provide the basis data for further microbiome studies on layer chickens. The 16s rRNA sequencing data has been evaluated in a wide range of analysis, and appears well thought through. The observations of the chicken microbiome during initial colonization and changes during the maturation period are highly interesting. Furthermore, the broader analysis linking the microbiome to functional pathways would make this paper stand out, but is somewhat missing from the discussion.
- The introduction, although well rounded on most aspects, would benefit from expanding further on how the microbiome can interact with metabolic pathways, as this would help to add weight to the study.
- Line 173-177: It is not clear if normalisation of the data was performed in order to remove bias for diversity analysis, this would be a necessary step.
- Line 191: How are you defining high quality, no degradation and good A260/A280 values? High quality here appears to be a subjective.
- Although a large portion of the results pertain to functional pathways, these are only loosely discussed. Perhaps an expansion of the effect a change in these metabolic pathways could be beneficial.
Author Response
Response to Reviewer 3 Comments
I firstly wanted to thank the authors for taking the time to alter and resubmit this article.
This revised and resubmitted article focusses on the microbial diversity found in the gut of layer chickens, in the first 57 days’ maturation. It was observed that the chicken gut microbiome stabilised around 50 days’ maturation, and there were distinct shifts in primary colonising species during this maturation period. The most prominent shift was from a primarily dominant lactobacillus colony, the microbiome adjusted to Bacteriodes, Odoribacter and Clostridales being more dominant. It is the authors hope that this work will be a reference point for further research on the microbiome of layer chickens.
Broad comments.
Point 1: This article is well organised and attempts to provide the basis data for further microbiome studies on layer chickens. The 16s rRNA sequencing data has been evaluated in a wide range of analysis, and appears well thought through. The observations of the chicken microbiome during initial colonization and changes during the maturation period are highly interesting. Furthermore, the broader analysis linking the microbiome to functional pathways would make this paper stand out, but is somewhat missing from the discussion.
Response 1: Thank you and we have reorganized and improved the discussion according to your suggestion. Please refer to line 448 to line 492.
Point 2: The introduction, although well rounded on most aspects, would benefit from expanding further on how the microbiome can interact with metabolic pathways, as this would help to add weight to the study.
Response 1: Thank you. We added the information according to your suggestion from line 109 to line 125.
Point 3: Line 173-177: It is not clear if normalisation of the data was performed in order to remove bias for diversity analysis, this would be a necessary step.
Response 3: Yes, normalization of the data was performed to remove bias for diversity analysis. We added the information from line 195 to line 197.
Point 4: Line 191: How are you defining high quality, no degradation and good A260/A280 values? High quality here appears to be a subjective.
Response 4: We believe that high-quality DNA first to have good A260/A280, no degradation, and the concentration of DNA meets the requirements of 16S rRNA gene sequencing. However, the high quality here means high quality sequences. We have revised the statement for better understanding. Line 230 to line232.
Point 5: Although a large portion of the results pertain to functional pathways, these are only loosely discussed. Perhaps an expansion of the effect a change in these metabolic pathways could be beneficial.
Response 5: Thank you. We added the information according to your suggestion. Please refer to lines 448-451 and lines 462-473.

Round 2
Reviewer 1 Report
The authors did not mention correct line numbers in their responses to comments number 28 and 29.
Author Response
Response to Reviewer 1 Comments
Point 1: The authors did not mention correct line numbers in their responses to comments number 28 and 29.
Response 1: Thank you. It was our mistake. We have revised the correct line numbers our responses to comments number 28 and 29 as follows:
Comments number 28: Lines 399-405: This study mainly focuses on the variation of gut microbiota of layer chickens with age but here the authors are taking about feeding probiotics and feeding methods. Please elaborate the connection.
Response 28: Thank you and we elaborate the connection according to your suggestion from line 469 to line 476.
Comments number 29: Lines 414-416: Please rewrite. There are certain studies available on the analysis of gut microbiota of chickens. One such example is "Cui, Y., Wang, Q., Liu, S., Sun, R., Zhou, Y., & Li, Y. (2017). Age-Related Variations in Intestinal Microflora of Free-Range and Caged Hens. Frontiers in microbiology.
Response 29: Thank you. We agree. We have revised the sentence according to your suggestion from line 489 and put this reference in line 417.

This manuscript is a resubmission of an earlier submission. The following is a list of the peer review reports and author responses from that submission.
Round 1
Reviewer 1 Report
The paper by Xiao et al analyzes the impact of spatial and temporal variation of microbiota on ammonia production in the intestines of laying hens. The authors attempted to establish an association between spatial and temporal variation of microbiota and ammonia emission in laying hens in the instant paper that is a very promising field of research. However, they did not mention any controls applied in the study. Further, the impact of feed on intestinal microbiota and gas production is of prime consideration in this type of studies but the authors did not provide any information about the composition of feed given to birds. Therefore, authors must improve methodology section.
Moreover, there are additional remarks and questions that authors should address.
- Lines 19 and 20: Please clarify and rephrase the sentence: “However, it is still not clear that spatial and temporal variation in the intestinal by ammonia-producing bacteria after birth of laying hens.”
- Lines 25, 26 and 33: Please replace the word “intestinal” with the word “intestines”.
- Line 26: Please remove the word “further”.
- Lines 30 and 31: Please rewrite the sentence “However, few studies have focused on spatial and temporal variation of the intestines by ammonia-producing bacteria.” May be authors mean that “Few studies have focused on establishing an association between the spatial temporal variation of intestinal bacteria and ammonia production in laying hens.”
- Line 40: Please clarify the sentence “However, the ammonia-producing bacterial advantage could not be maintained for a long time” and explain the ammonia-producing bacterial advantage in the context of this sentence.
- Line 47 and 48: Please rewrite the sentence “which is not conducive to the growth of the caecum and colon by ammonia-producing bacteria.” for proper understanding.
- Lines 26-27 and 48-49: There is repetition of same sentence “study provides a reference for further reducing ammonia emissions of laying hens by regulating intestinal microbes”.
- Lines 54-75 and 79-84: Please remove these lines and provide relevant information.
- Lines 95 and 96: Please elaborate the relationship between microbes and ammonia emission in laying hens.
- Lines 98-105: Please replace these lines with information relevant to effects of changes in microbiota on ammonia production.
- Lines 110: Please replace the words “newborn chickens” with the words “newly hatched chicks.”
- Lines 111-112: Please replace the words “With the continuous entry of the environmental microbiota, the intestinal microbiota” with “However, the microbiota grows very fast during first week and”.
- Lines 113-115: Please rewrite in a more readable way.
- Lines 158 and 159: Please remove these lines.
- Lines 388-404: Overall conclusion does not state any information about the association between changes in intestinal microbiota and ammonia production in laying hens.
Reviewer 2 Report
Functional microbial spatial and temporal variation associated with ammonia emission in the intestines of laying hens
Manuscript ID: animals-1005500
The present study deals with the reduction of ammonia emission from poultry. The authors investigated the content of different intestinal parts and made 16S rRNA sequencing to determine the composition of microbiota. The issue itself is of importance in veterinary medicine, but I see some major hindrances for publication.
Intestinal samples from birds were investigated until the age 57 days. With this age layers are still in their chicken age and do not produce eggs. Therefore, the title shall be changed accordingly. Also, this has to be changed in the whole manuscript: The authors have to use the term layer chickens. The fact that young birds were used in the present study is a certain hindrance to conclude what might happen in older layers (e.g. start of lay, peak of lay, end of lay) when certain hormonal changes might also have an influence on the gut microbiota. Another critical point is the feed. Also, information is needed if feed supplements or probiotics were provided. Were the birds treated within the period of investigations (e.g. antibiotics). In which housing system were the birds kept? These factors are known to have an impact on the composition of the gut microbiota. This issues need also critical reflection in discussion. The writing needs improvement.
Detailed comments:
Line 19-20: The sentence is not complete.
Line 20: The authors shall use after hatch instead of after birth.
Line 21: Which type of Lohmann layers (brown, white, etc.)? As there might also be differences in the composition of microbiota.
Line 20-25: Make 2 or 3 sentences – one long sentence is very confusing.
Line 22: The authors shall use after hatch instead of after birth
Line 26: in the intestine of laying hens instead of intestinal.
Line 32: Not the birds were analyzed but samples from them. Please change this accordingly.
Line 33: intestine instead of intestinal.
Line 40: This sentence is not understandable. Do the authors mean that the high level of ammonia-producing bacteria decreased and was gradually replaced by Lactobacillus species?
Line 94: use birds instead of laying hens.
Line 110: use newly hatched chickens instead of newborn chickens.
Line 116: by exploring instead of by explore.
Line 124: Name the husbandry system. The gut microbiota can also be influenced by the husbandry system used. E.g. coprophagia in cages vs. floor housing.
Line 131: Which feed did the chickens/pullets receive? Did the feed contain any supplements? Were probiotics used? Which bedding material was used?
Line 132: randomly instead of random.
Line 134: Which amounts of the contents from each intestinal part were taken for investigation? Were the samples pooled from the 6 birds?
Line 139-140: three different positions – but only two are named: the front and the middle.
Line 326: intestinal microbiota instead of intestine microbiota.
Line 328: Make an own sentence and remove spatial.
Line 329: This sentence needs modification. E.g. Here, using samples from duodenum, caecum and colon the spatial and temporal variation of the intestinal microbiota in layer chickens were systematically studied.
Line 331: It is not clear what the authors mean with: further explaining the loss of gaseous nitrogen.
Line 350: It is not clear what the authors mean with this sentence.
Line 352: use at hatch instead of at birth.
Line 356: Again, these birds were 50 days of age: they were not mature, and their intestinal microbiota was also not mature.
Line 359: newly hatched chicks instead of newborn chicks
Line 365: This sentence is not understandable.
Lines 374-377: This is not understandable.
Line 381-383: This is not understandable.
Line 383: The farm was a commercial one, and not an experimental farm.
Line 389: Change sentence to: In many studies different ammonia-producing bacteria…
Line 393: at hatch instead at birth
Line 396: birds are 50 days – no mature birds, no mature microbes.
Line 404: Which new ideas to these results provide? If the amount of ammonia-producing bacteria declines with the age of birds; I would conclude that ammonia emission will be no problem later on. But this conclusion is definitively contradictory to the real scenario worldwide.
Reviewer 3 Report
Functional microbial spatial and temporal variation associated with ammonia emission in the intestines of laying hens
Summary
This article focusses on the spatial and temporal variation of ammonia producing bacteria in the intestines of Lohmann laying hens. This was done through the use of 16S rRNA sequencing analysis and showed that the diversity in the intestinal microbiome increased with the age of the hens up until day 50 where it plateaued. It was also observed that at the start of the experiment the composition from each intestinal region was similar, with Escherichia and Clostridium as their main components. As the experiment continued, it was observed that replaced the ammonia-producing bacteria as the main component of the microbiome (Lactobacillus in the duodenum and Bacteriodes, Odoribacter, and Clostridiales vadin BB60 group in the caecum and colon). The author hopes that this work will provided a reference for further work on reducing ammonia emissions of laying hens through the regulation of intestinal microbes.
Broad Comments:
This article is well organised and attempts to shed light on the presence of ammonia producing bacteria in the intestines of laying hens. The 16s rRNA sequencing data appears well thought through, and could help to forward our understanding of the chicken microbiome especially in relation to ammonia production in laying communities. There are however, numerous grammatical errors, especially in the introduction, that lead to the article being hard to follow in places and as such would benefit from a broad review.
Specific comments
- The introduction, although well rounded on some aspects, would benefit from expanding further on how ammonia from laying hens contributes to the global production of nitrogenous substances, as this would help to add weight to the study.
- Line 66: It is stated that the use of nitrogen fertiliser increased from zero to 83 million tons between 1900 and 2000. Is this statement truly correct, or is this not that records are not present from before this, please clarify this statement.
- The author has stated that laying hens produce less in response to high ammonia level, how this truly impacts productivity would be an interesting addition to the introduction, leading to a better understanding of the urgency or necessity for laying facilities to look at reducing ammonia production.
- It is stated throughout the article that the author is looking at microbiota maturity, but this is not technically correct. Is it not the case that the author is looking at microbiota in relation to animal maturity? This statement throughout the article is therefore somewhat ambiguous for the reader, and leads to some difficulty in truly understanding what the author means here.
- Line 104: The line “Amino acids produce ammonia” should be altered to “Amino acids are converted to ammonia”.
- Although the information is present, the formatting of the material and methods seems out of order, with the Measuring of ammonia being sandwiched in between sample collection and the extraction methodology. I would recommend that the order be revised in order to flow better.
- As part of the sample collection methodology section, it is not clearly stated how many animals were sacrificed in total, and how they were sacrificed.
- Although there are a number of figures present in the article itself, the PCOA plot in Figure 2 needs deeper explanation of the data in order to justify its inclusion, and aid the reader in their understanding of the data presented.
- It is stated on Line 264 that “the gut microbiota had reached maturity”, this statement could again benefit from clarification that this is now a stable microbiome that no longer undergoing fluctuations in its composition. There needs to be a distinction made between the individual microbes and a microbiome as a whole, which is not currently clear in the article.
- In the discussion, the author briefly touches upon the fact that different intestinal regions have different physiological functions. This should be expanded upon to give the reader some understanding of how this can affect the microbiota.
Reviewer 4 Report
The authors reported the presence of ammonia-producing bacterial in intestines might be the main reason for animal ammonia production. For that, the authors have been investigated the spatial and temporal variation of the intestines microbiota using 16S rRNA sequencing. The outcome reveals that the microbial community structure in the duodenum, caecum, and colon increased with age and tended to be stable when the hens were 50 days of age. However, the authors missed providing environmental and sample controls during their sampling, and sample processing makes questionable and hard to correlate the microbial community when all hens were placed in the same places. The authors missed how the feces microbiota will influence the hens intestine after a few days. I would suggest the author provide detailed information about why they have not included controls and environmental microbiota.
Authors focused only on the animal ammonia intestine microbiota and missed to provide data related to the environmental influence of the feces when all animals have been put together in a cage. The authors should have explained why they have not included environmental control in their experiment.
The authors provided the b-diversity of the microbiota individual sample type and not explained it well. However, the author has missed how all three of b-diversity might look in the same PCoA plot. I would suggest them to provide detailed information and comparison of three sites b-diversity as one figure (PCoA plot)
Figure 3 results authors should provide detailed comparison within and between sampling sites
Figure 4 results unable to understand authors should provide detailed comparison and correlation between age and microbiota.
Round 2
Reviewer 1 Report
Although the authors made some improvements in the article, however, the article still needs significant improvement. Methodology applied must be improved. Authors must state the procedure of selection of hens for sampling. For example, they mentioned that ammonia concentration was different in different sections of the animal house but they did not mention whether laying hens used for sampling were selected from different sections of the house (front, rear and middle).
Further, variations of ammonia producing bacteria in the intestines of laying hens from different sections of the animal house were not mentioned. These are important aspects to corelate variations of ammonia producing bacteria with ammonia production.
Reviewer 2 Report
I reviewed the revised version of the manuscript "Functional microbial spatial and temporal variation associated with ammonia emission in the intestines of laying hens".
The authors did an extensive and careful corection of their manuscript. I just have some minor comments. I just realized that the title has to be changed as follows: Functional microbial spatial and temporal variation in the intestine of layer chickens associated with ammonia emission.
Line 21: The sentence needs to be adjusted: However, there are no reports on the spatial...
Lines 24-27: The sentence is very long. I suggest to change it accordingly: Contents of duodenum, caecum and colon of white Lohmann layer chcikens from the same breeding house were collected on fourteen time points (0,1,..). By applying 16s rRNA sequencing the composition of ammonia-producing bacteria was determined.
Reviewer 4 Report
I would suggest authors should include environmental controls even though they have maintained the cage routinely. The environment will influence more on the dynamics of the microbial diversity in the given sample as similar to the food and neighbors. So it is necessary to have controls (Environmental controls, sample positive and negative controls to check the NGS).
